# Polyolefin Blends with Selectively Crosslinked Disperse Phase Based on Silane-Modified Polyethylene

**DOI:** 10.3390/polym15244692

**Published:** 2023-12-13

**Authors:** Markus Gahleitner, Tung Pham, Doris Machl

**Affiliations:** 1Borealis Polyolefine GmbH, Innovation Headquarters, St. Peterstr. 25, 4021 Linz, Austria; doris.machl@borealisgroup.com; 2Research Institute of Textile Chemistry and Textile Physics, University of Innsbruck, Hoechsterstrasse 73, 6850 Dornbirn, Austria; tung.pham@uibk.ac.at

**Keywords:** polypropylene, polyethylene, multiphase, crosslinking, morphology, mechanics, rheology

## Abstract

Polypropylene-based multiphase compositions with a disperse elastomer phase provide superior impact strength. Making this property indifferent to processing steps requires stabilization of the morphology of these materials. Various approaches have been tested over time, each of which shows limitations in terms of performance or applicability. Using polyethylene (PE) homo- and copolymers capable of silane-based crosslinking as modifiers was explored in the present study, which allows decoupling of the mixing and crosslinking processes. Commercial silane-copolymerized low-density PE (LD-PEX) from a high-pressure process and silane-grafted high-density PE (HD-PEX) were studied as impact modifiers for different types of PP copolymers, including non-modified reference PE grades, LDPE and HDPE. Blends based on ethylene–propylene random copolymers (PPR) and based on impact- (PPI) and random-impact (PPRI) copolymers show improvements of the stiffness–impact balance; however, to different degrees. While the absolute softest and most ductile compositions are achieved with the already soft PPRI copolymer base, the strongest relative effects are found for the PPR based blends. Modifiers with lower density are clearly superior in the toughening effect, with the LD-PEX including acrylate as second comonomer sticking out due to its glass transition around −40 °C. The impact strength improvement found in most compositions (except at very high content) results, however, not from the expected phase stabilization. For comparable systems, particle sizes are normally higher with crosslinking, probably because the process already starts during mixing. Thermoplastic processability could be retained in all cases, but the drop in melt flow rate limits the practical applicability of such systems.

## 1. Introduction

Polyolefins, and especially isotactic polypropylene (iPP), have gained wide acceptance in multiple areas over the last 60 years, resulting in a global production volume of about 75 million tons in 2020. Copolymers, blends, and composites based on iPP, but characterized by a multiphase structure with crystalline matrix and amorphous or low crystallinity particles (commonly termed ‘elastomeric’), have been essential for this success, especially for technical applications like the automotive or appliance areas. The structure is decisive for combining stiffness and toughness, but also for extending application temperature range into the region below the glass transition (Tg) of the base polymer, which is ~0 °C for iPP homopolymers [1,2,3].

From the early days of pure compounding through the development of heterophasic copolymers up to present developments, the issue of processing effects on phase structure has remained. These morphology changes, including particle orientation and breakup, but sometimes also agglomeration, affect the final performance in cases of fully thermoplastic systems [4,5,6,7]. The changes concern mechanics, especially impact strength and ductility, but also shrinkage and thermal expansion (in terms of magnitude and anisotropy), and surface quality. With the increasing complexity of part geometry, this issue has become more relevant over time.

Numerous approaches for stabilizing the elastomeric particles have been developed and studied through the years. A classical approach is vulcanized (i.e., classically sulfur-crosslinked) natural rubber blends, evaluated, for example, by Varghese et al. [8] and Passador et al. [9]. While vulcanization is an established process, which can be controlled well in terms of crosslinking density, the development work has mostly been limited to compositions with a majority of elastomer phases. Applicability of this concept is, moreover, limited by the inherent smell from the sulfur components.

More flexible in terms of composition and crosslinking system are iPP blends with ethylene–propylene-diene (EPDM) elastomers. Conventional vulcanization of such compositions has, for example, been explored by Sengupta and Konar [10] and Naskar [11]. An alternative approach, the use of imide as a selective crosslinker for EPDM, has been studied by Inoue [12] and Ishikawa et al. [13]. Studies using the combination of phenolic resin and tin chloride, SnCl_2_, have been presented by Ellul et al. [14] and l’Abee et al. [15]. Finally, the use of radiation for EPDM crosslinking has been evaluated by Zaharescu et al. [16,17]; however, finding concurring degradation effects in the iPP matrix. The latter phenomenon resulting from the different reaction of ethylene(C2)- and propylene(C3)-based chains to radicals is a general issue of all radical-based crosslinking processes. The respective papers cover a wide range of composition and particle size, mostly again showing a clear mechanical advantage of the crosslinking. Except for the radiation approach, however, all systems are known to have emission problems limiting applicability in sensitive areas like automotive interior.

Radical-based crosslinking, normally using organic peroxides (POX) as initiators, have been evaluated for iPP blends with conventional ethylene–propylene (EPR) elastomers, i.e., without diene component, and with various types of polyethylene (PE). The concept was shown to work for EPR alone [18] or in ternary blends with high density polyethylene (HDPE) [19], again with the limitation of parallel iPP matrix degradation. Moreover, this results in rather coarse blend morphologies and, consequently, limited mechanical performance. Pure PP-PE blends with POX normally give a combination of PE crosslinking and PP degradation. An expansion of this concept has been tested for low density polyethylene (LDPE) [20] and HDPE [21] through combination of POX with bi- or multifunctional crosslinking agents. Most relevant in terms of impact strength improvement are blends with very low-density PE plastomers like ethylene-hexene- or ethylene-octene plastomers [22,23]. Probably due to the increased reactivity to radicals of such systems, as a consequence of higher unsaturation content, POX alone has been shown to give acceptable results in these cases.

Reactively modified ethylene–propylene impact copolymers (ICPs), commonly also called EP-HECOs [2], are similar in reactivity to PP/PE blends, even if their structure is more complex and normally comprises three phases: A matrix being a PP homopolymer or random copolymer, an amorphous EP copolymer (EPC) and, especially at higher C2 content in the EPC, crystalline inclusions of composition similar to a linear low density polyethylene (LLDPE) with C3 as comonomer and analogous behavior. They also show a similar mix of degradation and crosslinking in the case of POX alone [24], requiring the addition of a crosslinking/coupling agent for efficient modification and disperse phase stabilization. This has been demonstrated in a post-polymerization process with the addition of butadiene [25], but also directly in polymerization through copolymerization with non-conjugated dienes [25,26,27].

In most of these cases, no separation between the mixing process—determined by the compatibility and viscosity ratio between the matrix and disperse phase [2,28,29] and crosslinking—is possible. Of all approaches discussed above, only post-vulcanization is a notable exception here.

Silane-based crosslinking of PE homo- and copolymers [30,31,32,33,34,35,36] should allow this separation, as the crosslinking process can be performed on the melt-mixed compound or even on the molded article. For this, first an organosilane group like vinyl(trimethoxy)silane (VTMS) is added to the polymer either by copolymerization, which works only in case of the high-pressure LDPE process [30,31,35], or after polymerization by peroxide-initiated grafting. The latter has been shown to work for HDPE, linear low-density PE (LLDPE), PE-plastomers and EPR [32,33,34,35,36]. The shaping (or mixing) process is followed by crosslinking under elevated moisture and temperature with a crosslinking catalyst, which accelerates and controls the hydrolysis step of the crosslinking process. PE-silane copolymers are known to crosslink rather slowly and inhomogeneously.

Blends with PP have been studied, for example, in the group of Fritz [34], but only for specifically grafted PE plastomers and again, like in case of PP/EPDM compositions mentioned above [11,12,13], rather as thermoplastic elastomers resp. vulcanizates, i.e., with a majority of the softer phase.

In contrast to that, the work presented here has its focus on impact modification with limited amounts of disperse phase. For the present study, we have used conventional silane-copolymerized LD-PEX and silane-grafted HD-PEX as impact modifiers for different types of PP copolymers, applying the same design principles as for conventional PP compounds without crosslinking of the disperse phase [37]. Non-modified reference LDPE and HDPE grades were included to complete the picture.

## 2. Materials and Methods

Only commercial products of Borealis AG were used for the compounds, for which Table 1 presents an overview of the composition and properties. The four PP grades used as base polymers cover most relevant copolymer types with ethylene (C2), being two random copolymers with single-phase structure (PPR1 and PPR2 [38]), one impact or heterophasic copolymers with multi-phase structure (PPI [2]), and one random-impact or random-heterophasic copolymer (RAHECO). While the PPI has a PP homopolymer matrix, the PPRI combines a PPR matrix with copolymer (EPC) inclusions (PPRI [2,39]). In the case of the latter two, the xylene cold soluble (XCS) fraction roughly indicates the amount of disperse copolymer phase, while the melting point (Tm) from differential scanning calorimetry (DSC) reflects the comonomer content in the crystalline matrix.

XCS was determined in accordance with ISO 16152 [40] at 25 °C using a sample mass of 1 g and a drying temperature of 90 °C. DSC analysis was run according to ISO 11357/part 3/method C2 [41], determining the melting peak temperature (Tm) and melting enthalpy (Hm) for the PP and PE components as well as the crystallization peak temperature (Tc) with a TA Instruments Q200 differential scanning calorimeter on 5 mg samples. DSC was run in a heat/cool/heat cycle with a scan rate of 10 °C/min in the temperature range of −30 to +225 °C. Only one crystallization temperature (T_c,PP_) was observed and determined from the cooling step, while the melting temperatures (T_m,PP_ and T_m,PE_) and the related melting enthalpies (H_m,PP_ and H_m,PE_) were determined from the second heating step. The total C2 content was measured by Fourier transform infrared spectroscopy (FTIR) calibrated by 13C nuclear magnetic resonance (NMR) spectroscopy.

For modification, two silane-modified copolymers of different density were used, one being a long-chain branched LDPE copolymer from a high-pressure process also including acrylate [42,43] and the other one a grafted HDPE. In the case of both polymers, the applied silane was vinyl(trimethoxy)silane (VTMS), and its content as well as the acrylate content were determined by FTIR (see Table 1). For both types of silane-modified PEX, a crosslinking catalyst masterbatch, type CAT-MB50 of Borealis with dibutyltin dilaurate as a catalytically active substance was added at a relative concentration of 2.5% of the added PE component. As reference modifiers, a conventional LDPE (long-chain branched) and HDPE (linear) of comparable density and molecular weight, i.e., melt flow rate (MFR) as determined in accordance with ISO 1133, were used. The PEX/PE addition to the PPR results in two-phase blends, while for the PPI and the PPRI base three-phase blends are generated, with the amorphous copolymer EPC acting as compatibilizer between PP and PE.

All compounds were produced in the same way, on a ThermoPRISM TSE24 twin-screw extruder configured with two high intensity mixing segments. A temperature profile as commonly applied for PP compounding in the range of 190 to 220 °C was selected, screw speed was set at 50 rpm and throughput at 10 kg/h. The compositions were extruded through two circular dies of 3 mm diameter each into a water bath of ambient temperature with a residence time of at least 30 s for solidifying the melt strand, which was subsequently pelletized. This short water contact will not be sufficient to ensure crosslinking of the PEX components. In order to complete the process, but also to have identical conditions for all compounds, the resulting pellets were stored at 23 ± 2 °C and 50 ± 5% relative humidity for at least 96 h.

As initial characterization, MFR and the xylene hot insoluble (XHU) fraction of all compositions as well as the PP base polymers were determined. The MFR measurement was performed to ensure thermoplastic processability. XHU was determined to assess the relative degree of crosslinking of the PEX phase, which was calculated by relating XHU to the modifier amount. Values of XHU below 1 wt.-% are no clear indication of crosslinking and may result from additives or contamination. Thermal properties were determined by DSC as explained above, and in Table 2 and Table 3 the compositions and basic properties are presented. The results are split into those for the compounds based on random copolymers (two-phase systems with single-phase references, Table 2) and those based on impact- and random-impact copolymers (three-phase systems with three-phase references, Table 3). It should be noted that H_m,PE_ is only comparable within one modifier type, as LDPE and HDPE have different degrees of crystallinity. In addition, the PPI and PPRI base polymers already contain a very small fraction of crystallizable PE, as commonly found for such polymers [2,28].

Structural details of the compounds were explored by melt rheology, dynamic-mechanical testing (DMA) in solid state and transmission electron microscopy (TEM), the latter being applied for selected samples only.

The melt rheological measurements were carried out according to ISO 6721-1 [44] and 6721-10 [45], using an Anton Paar MCR 301 Rheometer (Anton Paar, Graz, Austria) equipped with a convection oven. The measurements were performed under nitrogen atmosphere to prevent oxidation and degradation. Parallel plate–plate geometry was used, with plates 25 mm in diameter. The frequency range was from 0.01 to 628 rad s-1 with a five-point averaging per decade. The applied strain was from 2–7% and the gap between the plates 1.3 mm. Storage and loss moduli G′(ω), G″(ω) were determined at 230 °C and the complex viscosity η*(ω) calculated from these.

Figure 1 represents the results for one part of the two-phase compounds, based on PPR2 with the higher MFR. The compounds with disperse PEX phase having a crosslinking degree of ~45% show an increasing gel-like behavior with increasing amount, with G′ being above G″ over the whole frequency range for 50 wt.-% LD-PEX. This seems to be in contrast to the thermoplastic processability of these compositions demonstrated by the MFR measurements and the specimen preparation for mechanical testing, but for partially crosslinked materials’ linear-viscoleastic properties cannot be expected to correspond to nonlinear ones. It should further be noted here that pure LD-PEX cannot be measured with conventional means, as the crosslinked material will slip on the plates.

The difference in complex viscosity at low frequencies (i.e., shear rates) is nearly two orders of magnitude between 25 wt.-% compounds with LD-PEX and non-crosslinked LDPE. This demonstrates the gel-like behavior and corresponding longer relaxation times of the crosslinked PE fraction. While the non-crosslinked PP matrix ensures processability, the effect is still observed at higher shear rates by a massive MFR difference (1.2 vs. 8.5 g/10 min).

DMA was performed according to ISO 6721-7 [46]. The measurements were performed in torsion mode on compression molded samples (40 × 10 × 1 mm^3^) between −100 °C and +150 °C with a heating rate of 2 °C/min and a frequency of 1 Hz. The storage modulus G′ was determined at +23 °C, and the various glass transitions as the median points of the peaks of the tangent of the loss angle, tan(δ). In case of the present compounds, we observed glass transitions in the range of −6 to 0 °C for PP, as well as around −50 to −30° C for the disperse copolymer phase (EPC) of the PPI and PPRI types [1,2] and the acrylate-containing LD-PEX, and below −100 °C for HDPE.

Figure 2 again gives a respective example for the concentration series of the three-phase compounds based on PPRI. The base polymer has a Tg of the EPC phase at −52 °C, which is shifted to slightly higher temperatures, but intensified in integral area by the LDPEX addition of 10 to 50 wt.-%. At the same time, the matrix-related peak of Tg(PP) shrinks, becoming rather flat for the 50 wt.-% compound. Moreover, the relative effect of pure LDPE for mobility at 25 wt.-% is significantly weaker.

Phase morphology analysis was based on transmission-electron microscopy (TEM), images being taken by combining the commonly used contrasting of ruthenium tetroxide (RuO_4_) [47] with cryo-ultramicrotomy. Samples were taken from the central part of a compression-molded DMA specimen, capturing images with a TEM Tecnai 12 (FEI) equipped with CCD camera (Gatan, Bioscan). In Figure 3, a comparison of two-phase compounds based on PPR1 and PPR2 with 25 wt.-% of LD-PEX and LDPE, resp., is presented. The matrix (PPR) phase is brighter due to higher density, while the disperse phase is contrasted stronger.

Mechanical performance of the compounds was tested according to standard procedures in tensile, flexural, impact and heat deflection tests. For all these, samples were injection molded following the general standard ISO 19069-2 [48]. Tensile tests were performed on at +23 °C according to ISO 527-2 [49] (cross head speed 1 mm/min for tensile modulus, 50 mm/min for other parameters) on type 1B dog bone specimens of 4 mm thickness. Tensile modulus, yield stress and strain at break were measured. Flexural modulus was determined at +23 °C in a 3-point-bending according to ISO 178 [50] on specimens of 80 × 10 × 4 mm^3^. Identical specimens were also used for Charpy notched impact strength determined according to ISO 179 1eA [51] (U-notch) at +23 °C and −20 °C, and for heat deflection temperature (HDT) determined according to ISO 75-B [52] (0.45 N/mm^2^ load).

## 3. Results and Discussions

The results will be discussed in line with the structures of the blends. In case of PPR matrices, two-phase blends as presented exemplarily in Figure 3 are found, meaning that there is a “hard” interface between PP and PE resp. PEX phase. For the PPI- and PPRI-based compositions the EPC fraction of the base polymer can act as a compatibilizer, resulting in a three-phase structure as observed before [37].

### 3.1. Two-Phase Compositions

Mechanical effects of LDPE blending into PP homopolymers has been studied by several groups before [20,53,54,55], partly because the long-chain branched LDPE has been used as a processing modifier before the development of PP grades with sufficient melt strength. The two polymers are immiscible in relevant concentrations and form matrix/inclusion morphologies at concentrations up to ~40 wt.-% of the minority blend partner, or a co-continuous morphology at roughly equal fractions [55]. Impact strength enhancement is found, but limited in extent.

In contrast to that, HDPE has been studied much less as blend partner for PP, because without compatibilization the mechanical effect is rather negative [21,56,57,58,59]. In earlier work in our group, changing the matrix of an HDPE/PP blend from PP homopolymer to PPR gives enhanced compatibility and consequently finer morphology. In the latter case blends up to 5 wt.-% of PE can be considered as homogeneous in the melt phase [58]. Still, the effect of HDPE on impact strength of the blends with PP is much worse than for PE plastomers with lower density [59], which are known to be efficient impact modifiers.

The latter is also confirmed in the present series of PPR-based compounds, the complete mechanical performance data of which are summarized in Table 4. While LDPE addition clearly improves the room temperature impact strength of PPR1, parallel to a significant reduction in modulus and heat resistances, HDPE addition reduces toughness. As its modulus is higher, however, it increases stiffness and even HDT. Moreover, the fact that at sub-ambient temperatures (−20 °C) the toughness effect of both PE types—LDPE and HDPE—is negative is in line with earlier results for conventional compounds [37,59].

When limiting the analysis to LDPE as a modifier, a clearly positive effect of crosslinking on impact strength, both at +23 and −20 °C, can be seen. This even extends to ductility as is expressed by strain at break. Figure 4 presents a typical stiffness–toughness diagram as preferred by plastics engineers for the two series. The limit between brittle and ductile behavior of PP-based compositions is commonly placed at a Charpy NIS level of 30–50 kJ/m^2^ [1,3,28], meaning that in both series, ductile behavior could be achieved with LD-PEX addition, but at different concentrations depending on the MFR resp. molecular weight of the base polymer, PPR1 and PPR2. As a look at the morphology images of Figure 3 shows, however, the positive effect of crosslinking is not a consequence of an improved morphology, like smaller soft-phase particle size [2,28,55]. For both base polymers, PPR1 (images a and b) and PPR2 (images c and d) the LD-PEX particles are bigger than the LDPE counterparts, meaning that the cohesive strength inside the disperse phase improved by the significant crosslinking degree of 43–58% (see Table 1) is obviously more decisive for mechanics than the morphology. This is in line with some of the earlier crosslinking studies discussed in chapter 1 [12,13,16,17], where, for example, EPDM crosslinking in a PP matrix was found to improve impact strength despite resulting in a coarser phase structure.

### 3.2. Three-Phase Compositions

For modifying PP impact copolymers, mostly HDPE [37,60,61,62,63,64] or PE plastomers [38,55,59,65] have been studied so far, probably because limited benefits were expected from LDPE. Notable exceptions are studies regarding rheology and especially melt strength of such compositions [66,67], as desired for foaming or thermoforming. In the present study, we limited the modification of the (stiffer) PPI with homopolymer matrix to HDPE resp. HD-PEX, and of the (softer) PPRI to LDPE resp. LD-PEX, the full mechanical data of which are summarized in Table 5.

Modifying impact copolymer by HDPE is a common practice in compound design, with reduced stress whitening of the resulting compositions being a key target [64]. It is taking advantage of the compatibilizing effect of the reactor-based amorphous ethylene–propylene copolymer (EPC) fraction, which was already demonstrated by D’Orazio et al. [60,68]. The combination of improved ambient-temperature impact strength with limited effect on modulus and sub-zero toughness shown before [1,37,63] is also found in the present study. Figure 5 again presents a typical stiffness–toughness diagram; however, comparing the two-phase series based on PPR1 to the three-phase series using the impact copolymer (PPI).

Both series show a strong positive effect of crosslinking on toughness at +23 °C, even in case of HDPE. Crosslinking of the PE inclusions (by 63%, see Table 2) here also causes a change from negative to positive impact strength effect at −20 °C (see data in Table 4 and Table 5). No morphology check was performed for these compositions, but the higher ductility of the PPI composition with non-crosslinked HDPE suggests smaller particles in that case, similar to the two-phase compositions.

More data are available for the LDPE/LD-PEX series based on the random-impact copolymer PPRI, which is characterized by a very fine phase structure with small EPC particles, resulting from good compatibility and good viscosity ratio between the matrix and the disperse phase [28,29,38]. This can be seen in image (a) of Figure 6, which combines TEM-based morphology images of the base polymer with those of 25 wt.-% blends with LD-PEX (b) and LDPE (c). The effect of crosslinking on particle size, which is clearly negative (i.e., increasing particle size) in the case of the PPR-based compositions, is less obvious here. Still, some agglomerates of EPC/LD-PEX particles are also seen in image (b). This difference may be related to a slightly lower degree of crosslinking achieved in the three-phase blends, which is in the range of ~37% (see Table 2). The combination of finer morphology and lower crosslinking density is also reflected in the rheology (see Figure 7; note that only for complex viscosity the *y*-axis scaling is identical to Figure 1). Even at 50 wt.-% of LD-PEX the “gel-like” behavior at low frequencies is much less expressed, despite an overall higher disperse phase content.

The toughness effect of crosslinking is clearly positive here as well, including the ductility parameter of elongation at break and with less drastic modulus effects resulting from the initially lower PPRI stiffness (see Figure 8). Room temperature impact reaches a level of 118 kJ/m^2^ for the PPRI-blend with 50 wt.-% of LD-PEX, which is at the upper method limit. This blend is highly ductile and shows no yield point in the tensile test. Two facts should be noted, however:

●For PPRI even non-crosslinked LDPE improves impact strength at +23 and −20 °C, in line with the changes in mobility observed in DMA (see Figure 2 and data in Table 4).●Despite the higher absolute toughness levels reached for PPRI-based blends with LD-PEX, the relative improvement is higher for PPR-base.

From an engineering perspective, and especially in terms of simplicity and processability, the finding regarding LDPE modification may be more relevant in practice than the crosslinking effect. As images (a) and (c) of Figure 6 show, the maximum soft-phase particle size is not increased by the LDPE addition, while additional smaller particles are generated. This is commonly considered beneficial for multiaxial impact strength, which was, however, not tested in the present study. PPRI/LD-PEX combinations with their outstanding toughness might find special applications, however.

## 4. Conclusions and Outlook

We explored a new approach for stabilizing the morphology of polypropylene-based multiphase compositions, using PE homo- and copolymers capable of silane-based crosslinking as modifiers. As the crosslinking takes place by a hydrolytic process on the final blends, this should theoretically allow decoupling the mixing and crosslinking steps [30,31,32,33,34,35,36]. In order to screen a wide composition range rapidly, commercial silane-copolymerized LD-PEX and silane-grafted HD-PEX were studied as impact modifiers for different types of single- and multiphase PP copolymers, reaching crosslinking degrees of 36–63%. Non-modified reference LDPE and HDPE grades were included in the study.

The two-phase blends based on ethylene–propylene random copolymers (PPR), as well as the three-phase blends based on impact- (PPI) and random-impact (PPRI) copolymers, show improvements of the stiffness–impact balance. Only LD-PEX and HD-PEX are capable of improving toughness of single-phase random copolymers in all respects, with non-crosslinked LDPE at least affecting ambient temperature impact positively. In both series ductile behavior could be achieved with LD-PEX addition, but at different concentrations depending on the MFR resp. molecular weight of the base polymer. This effect is also obvious when comparing Charpy NIS values at 23 °C for compositions with 25 wt.-% LD-PEX: 80 kJ/m^2^ for PPR1 with MFR 0.25 and 19.2 kJ/m^2^ for PPR2 with MFR 8.

The situation is similar in an impact copolymer with homopolymer matrix (PPI), but here the effect difference is between HDPE and HD-PEX. For the random-impact copolymer, both LDPE and LD-PEX modification gives positive effects, but to a different extent. The softest and most ductile compositions result from the already soft PPRI copolymer base, its ambient temperature impact strength of 118 kJ/m^2^ reaching the upper method limit. Despite this, the strongest relative improvements are possible for the PPR based blends. As seen for the difference between HDPE and PE plastomers before [49,65,69,70], PE modifiers with lower density are clearly superior in the toughening effect. Here, the LD-PEX, including acrylate as a second comonomer, is likely favored by its glass transition around −40 °C. Highly unusual for multiphase materials, even strain at break is increased from 200–300% for the PPR base polymers to nearly 600% for compounds with 25 wt.-% LD-PEX.

Taking all results together, one can conclude that the impact strength improvement found in most compositions does not result from the expected phase stabilization. For compositions comparable in terms of PP and PE component, particle sizes are mostly higher and at best comparable with crosslinking, probably because the process starts during mixing already. Thermoplastic processability could be retained in all cases, even if at very high LD-PEX content a “gel-like” behavior appears at low frequencies. Still, the drop in melt flow resulting from crosslinking rate limits the practical applicability of such systems, even if the ductility and toughness of the PPRI/LD-PEX combinations is outstanding. In terms of simplicity, the fact that simple LDPE modification of PPRI improves all toughness parameters may be more relevant in practice than the positive crosslinking effect.

## 5. Patents

The development work described here resulted in one patent application [71].

## Figures and Tables

**Figure 1 polymers-15-04692-f001:**
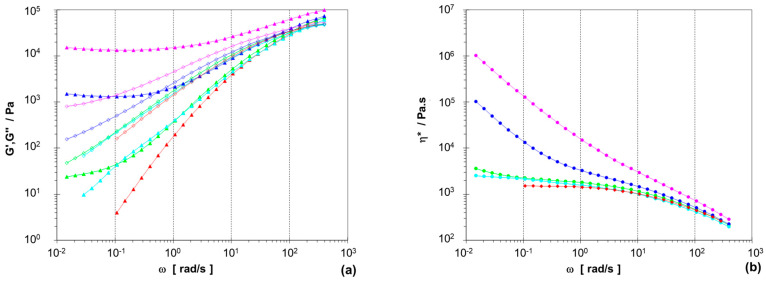
Rheological behavior (230 °C) of two-phase PPR2/LD-PEX series with different concentrations of modifier: 10 wt.-% (green, **–** ), 25 wt.-% (blue, **–** ) and 50 wt.-% (pink, **–** ), in comparison to base polymer PPR2 (red, **–** ) and PPR2/LDPE blend at 25 wt.-% (turquoise, **–** ); (**a**) storage modulus G′ (▲) and loss modulus G″ (◊), (**b**) complex viscosity η* (●).

**Figure 2 polymers-15-04692-f002:**
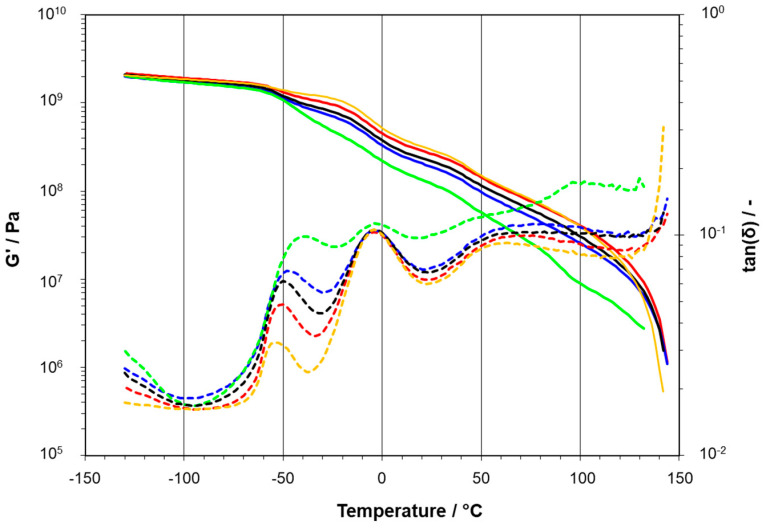
DMA results of three-phase PPRI/LD-PEX series with different concentrations of modifier: 10 wt.-% (red, **–** ), 25 wt.-% (blue, **–** ) and 50 wt.-% (green, **–** ), in comparison to base polymer (orange, **–** ) and PPRI/LDPE blend at 25 wt.-% (black, **–** ); solid lines represent storage modulus G′ and dashed lines loss angle tan(δ).

**Figure 3 polymers-15-04692-f003:**
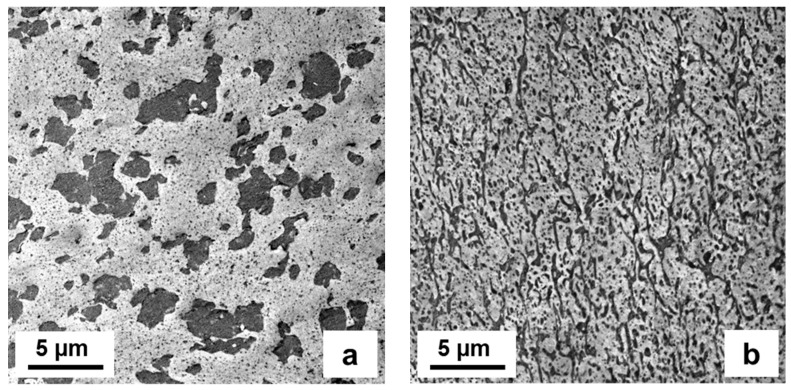
Phase morphology by TEM for two-phase compounds with 25 wt.-% modifier content; (**a**) PPR1/LD-PEX25, (**b**) PPR1/LDPE25, (**c**) PPR2/LD-PEX25 and (**d**) PPR2/LDPE25.

**Figure 4 polymers-15-04692-f004:**
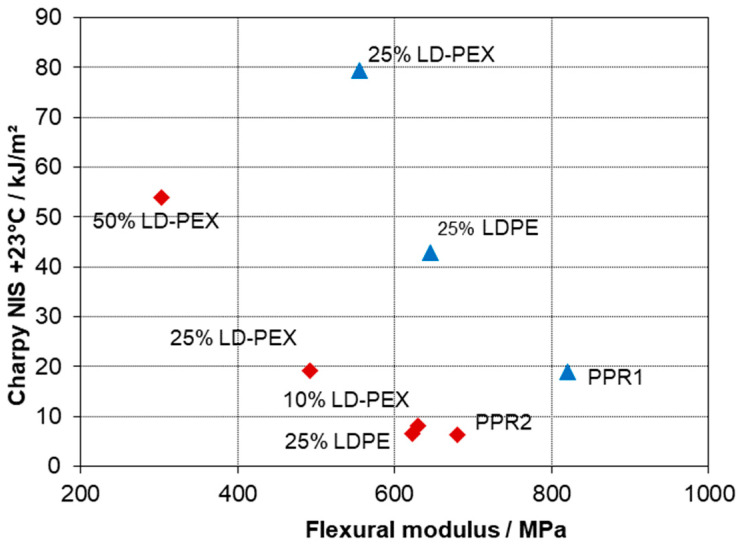
Stiffness–toughness balance of PPR base polymers and two-phase blends with LD-PEX resp. LDPE (series based on PPR1 blue, PPR2 red).

**Figure 5 polymers-15-04692-f005:**
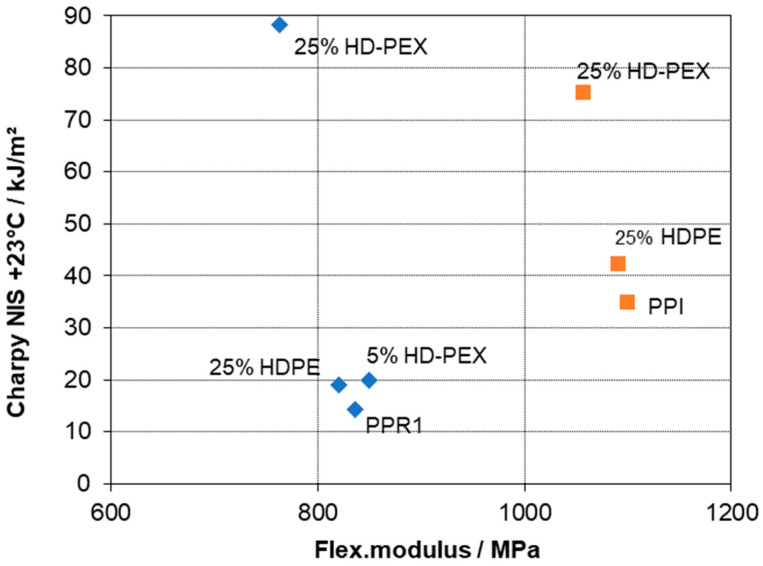
Stiffness–toughness balance of PPR1 and PPI base polymers and two- resp. three-phase blends with HD-PEX resp. HDPE (series based on PPR1 blue, PPI orange).

**Figure 6 polymers-15-04692-f006:**
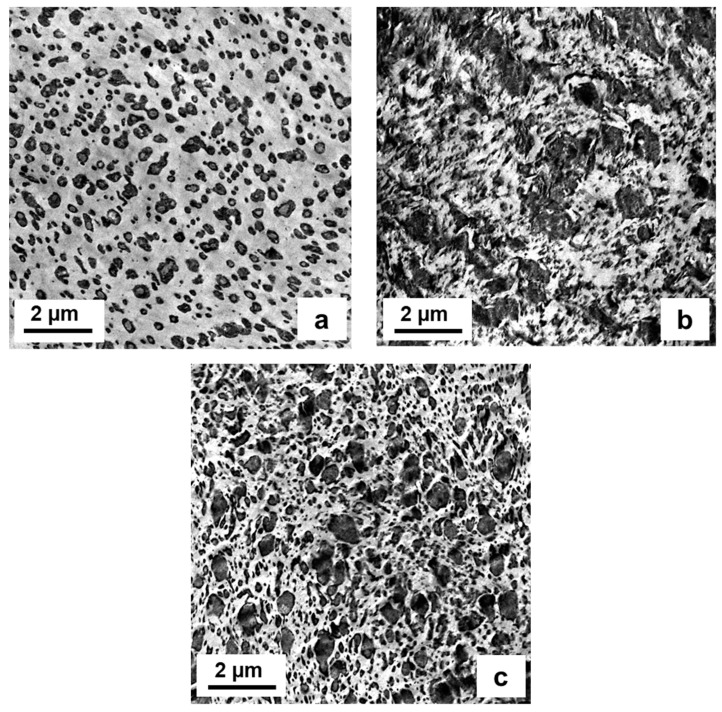
Phase morphology by TEM for (**a**) PPRI base polymer and three-phase compounds with 25 wt.-% modifier content, (**b**) PPRI/LD-PEX25 and (**c**) PPRI/LDPE25.

**Figure 7 polymers-15-04692-f007:**
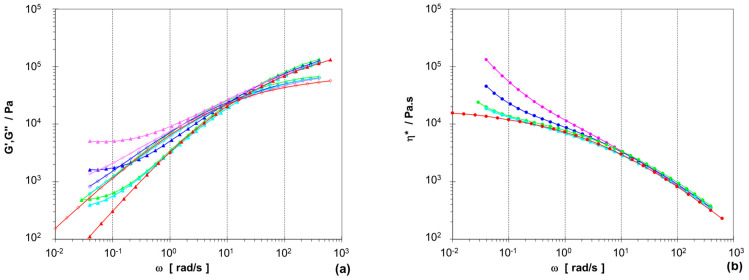
Rheological behavior (230 °C) of three-phase PPRI/LD-PEX series with different concentrations of modifier: 10 wt.-% (green, **–** ), 25 wt.-% (blue, **–** ) and 50 wt.-% (pink, **–** ), in comparison to base polymer PPR2 (red, **–** ) and PPR2/LDPE blend at 25 wt.-% (turquoise, **–** ); (**a**) storage modulus G′ (▲) and loss modulus G″ (◊), (**b**) complex viscosity η* (●). Note that only for (**a**) the y-scale (G′,G″) is identical to Figure 1 allowing direct comparison.

**Figure 8 polymers-15-04692-f008:**
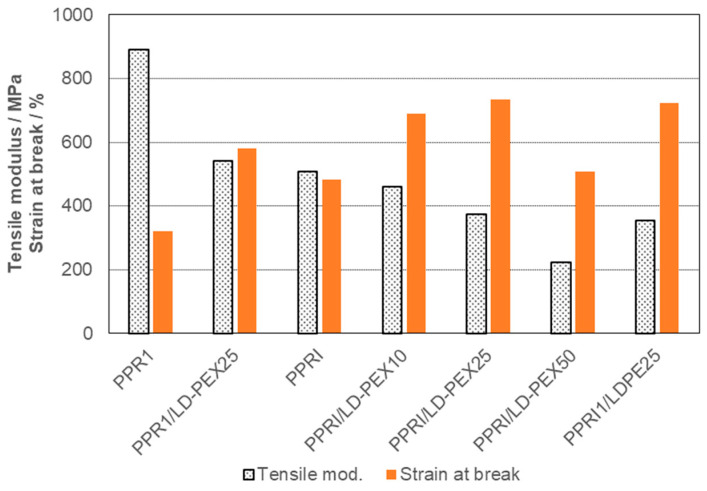
Modulus and ductility effects of LD-PEX resp LDPE modification of PPR1 (two-phase blend) and PPRI (three-phase blends).

**Table 1 polymers-15-04692-t001:** Compositions and properties of base polymers (PP) and modifiers (PE).

Type	Density	MFR ^1^	C2 Total	XCS	Tm
	kg/m^3^	g/10 min	wt.-%	wt.-%	°C
PPR1	905	0.25	3.9	7.0	143
PPR2	905	8.0	4.8	6.8	139
PPI	900	1.10	7.5	16.5	165
PPRI	900	1.00	9.5	23.0	143
**Type**	**Density**	**MFR ^1^**	**VTMS ^2^**	**Acrylate**	**Tm**
	**kg/m^3^**	**g/10 min**	**wt.-%**	**wt.-%**	**°C**
LD-PEX	927	5.0	1.75	18	93
LDPE	929	3.0	0.00	0	115
HD-PEX	942	1.00	2.00	0	128
HDPE	954	4.0	0.00	0	132

^1^ at 2.16 kg load, 230 °C for PP and 190 °C for PE; ^2^ Vinyl(trimethoxy)silane content, copolymerized in case of LD-PEX resp. grafted in case of HD-PEX.

**Table 2 polymers-15-04692-t002:** Compositions and basic properties of compounds based on random copolymers (PPR).

Compound	Base	Modifier		MFR ^1^	XHU	X-Linked ^2^	DSC				
							T_m,PE_	H_m,PE_	T_m,PP_	H_m,PP_	T_c,PP_
	Type	Type	wt.-%	g/10 min	wt.-%	%	°C	J/g	°C	J/g	°C
PPR1	PPR1	-	0	0.25	0.0	0	-	-	143	95.6	106
PPR1/LD-PEX25	PPR1	LD-PEX	25	0.15	14.5	58	91	7.1	142	66.7	104
PPR1/LDPE25	PPR1	LDPE	25	0.60	0.2	0.9	114	51.0	142	42.2	102
PPR1/HD-PEX05	PPR1	HD-PEX	5	0.20	2.5	51	127	4.9	143	67.2	110
PPR1/HD-PEX25	PPR1	HD-PEX	25	0.15	17.7	59	128	47.1	144	36.7	114
PPR1/HDPE25	PPR1	HDPE	25	0.80	0.1	0	131	64.8	144	44.3	115
PPR2	PPR2	-	0	8.0	0.0	0	-	-	139	86.0	99
PPR2/LD-PEX10	PPR2	LD-PEX	10	6.5	4.9	49	92	0.6	140	81.1	99
PPR2/LD-PEX25	PPR2	LD-PEX	25	1.2	10.8	43	93	5.2	139	69.1	99
PPR2/LD-PEX50	PPR2	LD-PEX	50	0.20	22.2	44	93	41.8	138	36.7	98
PPR2/LDPE25	PPR2	LDPE	25	8.5	0.2	0.8	114	41.1	139	63.6	100

^1^ at 2.16 kg load and 230 °C; ^2^ related to disperse (PE) phase.

**Table 3 polymers-15-04692-t003:** Compositions and basic properties of compounds based on impact (PPI) and random-impact (PPRI) copolymers.

Compound	Base	Modifier		MFR ^1^	XHU	X-Linked ^2^	DSC				
							T_m,PE_	H_m,PE_	T_m,PP_	H_m,PP_	T_c,PP_
	Type	Type	wt.-%	g/10 min	wt.-%	%	°C	J/g	°C	J/g	°C
PPI	PPI	-	0	1.1	0.0	0	123	0.8	166	89.6	118
PPI/HD-PEX25	PPI	HD-PEX	25	0.80	15.7	63	125	38.4	166	60.9	119
PPI/HDPE25	PPI	HDPE	25	1.8	0.2	0.8	128	31.6	166	72.1	117
PPRI	PPRI	-	0	1.0	0.0	0	111	0.8	143	65.3	102
PPRI/LD-PEX10	PPRI	LD-PEX	10	0.50	3.6	36	92	1.2	144	54.4.	125
PPRI/LD-PEX25	PPRI	LD-PEX	25	0.13	9.2	37	92	8.2	144	43.1	126
PPRI/LD-PEX50	PPRI	LD-PEX	50	0.05	19.1	38	92	22.9	143	12.2	126
PPRI/LDPE25	PPRI	LDPE	25	1.1	0.2	0.8	112	6.9	143	56.6	117

^1^ at 2.16 kg load and 230 °C; ^2^ related to disperse (PE) phase.

**Table 4 polymers-15-04692-t004:** Mechanical properties of compounds based on random copolymers (PPR).

Compound	Tensile Test			Flexural	Charpy NIS ISO 179 1eA		HDT
	Modulus	Yield Stress	Strain at Break	Modulus	+23 °C	−20 °C	ISO75B
	MPa	MPa	wt.-%	MPa	kJ/m^2^	°C	°C
PPR1	890	24.5	320	820	19.0	2.0	65
PPR1/LD-PEX25	540	18.9	580	555	80	5.8	59
PPR1/LDPE25	635	20.5	333	646	43	1.2	61
PPR1/HD-PEX05	841	25.9	379	849	19.9	1.5	70
PPR1/HD-PEX25	751	24.1	465	763	88	3.5	68
PPR1/HDPE25	854	24.5	374	836	14.3	1.1	69
PPR2	730	23.0	212	680	6.2	1.2	60
PPR2/LD-PEX10	603	19.2	582	630	8.2	1.9	59
PPR2/LD-PEX25	482	16.1	724	492	19.2	2.0	55
PPR2/LD-PEX50	310	11.9	531	302	54	4.9	48
PPR2/LDPE25	624	19.4	383	622	6.5	1.0	59

**Table 5 polymers-15-04692-t005:** Mechanical properties of compounds based on impact copolymer (PPI) and random-impact copolymer (PPRI).

Compound	Tensile Test			Flexural	Charpy NIS ISO 179 1eA		HDT
	Modulus	Yield Stress	Strain at Break	Modulus	+23 °C	−20 °C	ISO75B
	MPa	MPa	wt.-%	MPa	kJ/m^2^	°C	°C
PPI	1190	22.1	105	1100	35	3.8	73
PPI/HD-PEX25	1068	25.3	129	1057	75	5.0	74
PPI/HDPE25	1164	25.9	370	1091	42	3.6	76
PPRI	508	16.6	484	476	92	8.4	53
PPRI/LD-PEX10	460	14.4	689	412	94	9.2	51
PPRI/LD-PEX25	374	12.5	734	335	95	15.1	49
PPRI/LD-PEX50	222	-^1^	507	208	118	76	45
PPRI/LDPE25	355	12.4	724	315	93	41	48

^1^ no yield point detected.

## Data Availability

All analytical and mechanical data are listed in Tables, while full melt rheology and DMA data can be supplied by the authors upon request.

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
