# Peer review of "Polyolefin Blends with Selectively Crosslinked Disperse Phase Based on Silane-Modified Polyethylene"

_polymers, 2023, doi:10.3390/polym15244692_

Round 1
Reviewer 1 Report
Comments and Suggestions for Authors
This work reports a comprehensive study on the approach for stabilizing the morphology of PP-based multiphase composites. This work is well written. Some strengths are as follows: This topic is interesting and original, and it is applicable to the industrial developments of functional polymers. The experimental setup is well-studied. Moreover, the data analyses and explanations are impressive. The qualities and quantities of results are within the standard of the articles published in Polymers. The discussions and conclusions are consistent with the evidence and arguments and address the main research objectives. The references are appropriate. Moreover, the outlook for future research and development is given. Only one minor point that needs to be fixed is that the authors should provide the DSC curves and stress-strain curves for mechanical testing in the manuscript, not just the values. Apart from that, I have no further concerns. Therefore, I strongly recommend publishing this article in Polymers.
P.S. It is a nice piece of work.
Author Response
We thank the reviewer for the nice comments. Unfortunately, however, we are not in the position to add DSC and/or tensile curves, as the respective data are too old and have already been deleted from our storage system.

Reviewer 2 Report
Comments and Suggestions for Authors
The manuscript related to “Polyolefin blends with selectively crosslinked disperse phase based on silane-modified polyethylene” has been reviewed. I appreciate the author the idea of this work that is useful for academia as well as the paper industry. However, some discussions seem inadequate for good understanding considering that publication should make some changes to improve the quality of the manuscript.
1. For the present study we have used conventional silane-copolymerized LDPE and silane-grafted HDPE as impact modifiers for different types of PP polymers. Should need to mention the Wt % percentage of silane and structure (Line 113-114).
2. Authors need to add crosslinking density of modified polyethylene.
3. In the whole article abbreviations or terms are confusing and need to be systematic such as HDPE addition reduces toughness but increases stiffness and Heat deflection temperature (HDT).
4. The difference in complex viscosity at low frequencies (i.e. shear rates) is nearly two orders of magnitude between 25 wt.-% compounds with LD-PEX and non-crosslinked LDPE, still corresponding at higher shear rates to a massive MFR difference (1.2 vs. 8.5 g / 10 min). can authors explain this?
5. This is in line with some of the earlier crosslinking studies discussed in Chapter 1 [12,13,16,17]. Make clear this sentence. 6. Some spacing and formatting errors, The measurements were done in torsion mode on compression molded samples (40x10x1 mm3), At the same time, the matrix-related peak of Tg(PP), withRuO4 (Line 208, 218, 226).
7. Authors should mention quantitative results in the conclusion.
Comments on the Quality of English Languageattached file
Author Response
We thank the reviewer for the constructive comments and suggestions.
- For the present study we have used conventional silane-copolymerized LDPE and silane-grafted HDPE as impact modifiers for different types of PP polymers. Should need to mention the Wt % percentage of silane and structure (Line 113-114).
The amount of silane being vinyl(trimethoxy)silane (VTMS) is listed in Table 1, the legend of which has been amended.
- Authors need to add crosslinking density of modified polyethylene.
The degree of crosslinking of the PEX fractions is listed in Tables 2 and 3, its calculation from the XHU fraction being explained in the paragraph between the tables.
- In the whole article abbreviations or terms are confusing and need to be systematic such as HDPE addition reduces toughness but increases stiffness and Heat deflection temperature (HDT).
Thanks for the comments – some parts were really confusing. The whole paper has been revised accordingly, and abbreviations harmonized.
- The difference in complex viscosity at low frequencies (i.e. shear rates) is nearly two orders of magnitude between 25 wt.-% compounds with LD-PEX and non-crosslinked LDPE, still corresponding at higher shear rates to a massive MFR difference (1.2 vs. 8.5 g / 10 min). can authors explain this?
The respective explanation has been expanded.
- This is in line with some of the earlier crosslinking studies discussed in Chapter 1 [12,13,16,17]. Make clear this sentence.
The explanation has been expanded – it mostly relates to PP/EPDM systems.
- Some spacing and formatting errors, The measurements were done in torsion mode on compression molded samples (40x10x1 mm3), At the same time, the matrix-related peak of Tg(PP), withRuO4 (Line 208, 218, 226).
As stated above, the whole paper has been revised.
- Authors should mention quantitative results in the conclusion.
The conclusions have been expanded, also mentioning some quantitative highlights.

Reviewer 3 Report
Comments and Suggestions for Authors
The manuscript deals with important and also interesting topic. There is evident novelty. The obtained data are bringing the advance into the studied subject. Methods are well described and easy to follow. The manuscript is very well written without any evident issues. Conclusions section is "making sense". Just very few minor points have to be corrected:
1. line 208: use symbol × instead of letter x and use appropriate superscript. Thus, "40x10x1 mm3" has to be rewritten to 40×10×1 mm3
2. Fig. 7 b): I recommend the Y axis range to 1E+06 only.
All in all, the manuscript is of enough interest and should be accepted after above mentioned corrections.
Author Response
Thanks for the helpful correction points.
- line 208: use symbol × instead of letter x and use appropriate superscript. Thus, "40x10x1 mm3" has to be rewritten to 40×10×1 mm3
This has been corrected.
- Fig. 7 b): I recommend the Y axis range to 1E+06 only.
This has been corrected, adding a respective comment in the legend that only for Figure 7 (a) the y-scale (G’,G’’) is identical to Figure 1 allowing direct comparison.
